# Domain Generalization without Excess Empirical Risk

**Ozan Sener**
Apple

**Vladlen Koltun**
Apple

## Abstract

Given data from diverse sets of distinct distributions, domain generalization aims to learn models that generalize to unseen distributions. A common approach is designing a data-driven surrogate penalty to capture generalization and minimize the empirical risk jointly with the penalty. We argue that a significant failure mode of this recipe is an excess risk due to an erroneous penalty or hardness in joint optimization. We present an approach that eliminates this problem. Instead of jointly minimizing empirical risk with the penalty, we minimize the penalty under the constraint of optimality of the empirical risk. This change guarantees that the domain generalization penalty cannot impair optimization of the empirical risk, i.e., in-distribution performance. To solve the proposed optimization problem, we demonstrate an exciting connection to rate-distortion theory and utilize its tools to design an efficient method. Our approach can be applied to any penalty-based domain generalization method, and we demonstrate its effectiveness by applying it to three examplar methods from the literature, showing significant improvements.

## 1 Introduction

Empirical risk minimization (ERM) is the workhorse of machine learning. In this setting, a learner collects a set of data, assumed to be independent and identically sampled from a particular distribution of interest. An empirical risk over these samples is later minimized, hoping that the resulting model will generalize to future unseen data from the same distribution. Although ERM is ubiquitous, it is brittle to minor shifts in the distribution, and the resulting models suffer a significant performance drop when evaluated beyond their training distributions. Examples of such distribution shifts include different hospitals for medical imaging [49], camera types for perception models [8], and time periods for natural language models [20]. The problem of learning models which generalize under such distribution shifts is studied in the literature under the rubric of *domain generalization*.

Given a collection of domains (data distributions), domain generalization aims to find models that generalize to unseen domains. Although the domain generalization literature is rich, many model-free methods follow a common recipe. A surrogate penalty that encapsulates *generalization* is designed and a joint optimization problem of minimizing the empirical risk with the surrogate penalty is solved. Typical such penalties are invariance to an adversarial domain classifier [17, 42], invariance of an optimal predictor [5], and uniformity of risk [35]. Although these approaches are technically sound, their effectiveness has been challenged by recent empirical benchmarks [23, 18].

Joint optimization of the empirical risk and penalty is a sound method since the error on the unseen distribution can be bounded as the combination of an error on the training distributions (empirical risk) and the gap between performance on training and unseen distributions (generalization gap). Although domain generalization literature focuses mainly on controlling the generalization gap, we hypothesize that this joint optimization can increase empirical risk, effectively hurting performance on both training and unseen distributions. In other words, the lacking empirical performance of existing algorithms might be because they fail to optimize the empirical risk (i.e., in-distribution performance) even though they learn to control generalize gap. Consider the extensive evaluation conducted on the WILDS benchmark [23]. ERM yields *higher* out-of-distribution performance than dedicated domain

36th Conference on Neural Information Processing Systems (NeurIPS 2022).

generalization methods on most problems. In contrast, when we look at the generalization gap (i.e., the difference between in-distribution and out-of-distribution performance), domain generalization methods have a smaller gap than ERM on most problems. This supports our hypothesis that existing algorithms might be creating too much excess empirical risk. Thus the motivating question of our work: *Can we perform joint optimization without causing any excess empirical risk?*

Solving joint optimization without sacrificing empirical risk is not generally possible unless surrogate penalty term and empirical risk align perfectly. Instead, we formulate an optimization problem with the explicit constraint of no excess empirical risk. In other words, we propose to minimize the surrogate penalty only if it is not hurting the empirical risk. Formally, instead of joint minimization of empirical risk and surrogate penalty, we minimize the surrogate penalty under the constraint of optimality of the empirical risk.

To design an algorithm for the proposed optimization problem, we start with searching for a formal definition of the optimality of the empirical risk. Since the applications of interest are non-convex, this can only be local optimality. Hence, the constraint we consider is convergence to a stationary point of the empirical risk. We show that biased gradient descent converges to a stationary point as long as the bias is bounded and decreases through optimization. Using this result, we design an iterative algorithm that seeks the update minimizing the surrogate penalty under the constraint of bounded deviation from the empirical gradient. This construction is closely related to the rate-distortion problem from information theory [13], and we draw on this vast literature to design an algorithm. Specifically, we propose a Blahut-Arimoto-style method [11, 4] to optimize the proposed objective.

Our method only considers the optimization aspect of the domain generalization problem and can be applied to any penalty-based model with minor modifications. We perform our primary empirical evaluation on the CORAL model due to its simplicity [42]. We evaluate on the large-scale domain generalization problems in the WILDS benchmark of real distribution shifts [23]. Our evaluation using the image, formal language, natural language, and graph modalities suggests that our method significantly improves performance. To further validate the generality of our method, we apply it to the FISH model [39] and VRex [25] showing similar gains on WILDS [23] and DomainBed [18].

## 2 Method

### 2.1 Formulation and the Preliminaries

In this section, we introduce our notation and present a unified treatment of existing penalty based domain generalization methods.

We are interested in a learning problem in the space of $\mathcal{X} \times \mathcal{Y}$, where $\mathcal{X}$ is the space of inputs and $\mathcal{Y}$ is the space of labels. In domain generalization, a collection of different domains $\hat{\pi} = \{\pi_1, \ldots, \pi_M\}$ as well as an independent and identical (iid.) dataset for each domain are given as $\{x_j^i, y_j^i\} \sim \mathcal{P}^{\pi_j}$ for domain $j$ with $N_j$ samples. Consider the union of these datasets as the training set of domain generalization in the form of tuples $\{x, e, y\}$ such that $x$ is the input, $e$ is the domain id, and $y$ is the label. We further consider a parametric function family $h(\cdot; \theta) \colon \mathcal{X} \to \mathcal{Y}$ and a loss function $l(\cdot, \cdot) \colon \mathcal{Y} \times \mathcal{Y} \to \mathbb{R}^+$. Throughout our discussion, we denote random variables with capital letters (e.g. $X, E, Y$) and their realizations with lower-case letters (e.g. $x, e, y$).

We are specifically interested in penalty based domain generalization methods where empirical risk and domain generalization penalty are jointly minimized. Each model defines a penalty following a different set of assumptions. For example, penalty is defined in terms of second-order statistics for CORAL [42] and in terms of sub-optimality of the domain-specific classifier for IRM [5]. Since we are interested in the optimization of these models, not their construction, we treat `Penalty` as a generic entity and consider the optimization problem defined by these methods. These methods aim to solve the following joint optimization problem:

$$\min_{\theta} \quad \mathcal{L}(\theta; \{x^i, e^i, y^i\}) + \texttt{Penalty}(\theta; \{x^i, e^i, y^i\}). \tag{1}$$

where $\mathcal{L}(\theta; \{x^i, e^i, y^i\})$ is the empirical risk defined as $\mathcal{L}(\theta) \triangleq \frac{1}{M} \sum_{j=1}^{M} \frac{1}{N_j} \sum_{i=1}^{N_j} l(h(x_j^i; \theta), y_j^i)$. Although, the penalty term is typically weighted, weighting can be considered as part of its definition.

## 2.2   Optimizing Domain Generalization Objectives

The optimization problem defined in (1) is a sound proxy in order to optimize the risk for the unseen domains. Consider an unseen distribution $\pi_\star$, its corresponding population risk can be bounded using the following straightforward decomposition;

$$\underbrace{\mathbb{E}_{x,y\sim\pi_\star}\left[l(h(x;\theta),y)\right]}_{\text{Risk on Unseen Domains}} \leq \underbrace{\mathbb{E}_{x,y\sim\hat{\pi}}\left[l(h(x;\theta),y)\right]}_{\text{Risk on Seen Domains}} + \underbrace{\left|\mathbb{E}_{x,y\sim\pi_\star}\left[l(h(x;\theta),y)\right] - \mathbb{E}_{x,y\sim\hat{\pi}}\left[l(h(x;\theta),y)\right]\right|}_{\text{Generalization Error}}.$$

(2)

Following this decomposition, the optimization problem in (1) is sensible since the empirical risk $\left(\mathcal{L}(\theta;\{x^i,e^i,y^i\})\right)$ approximates the risk on seen domains and a the penalty term $\left(\texttt{Penalty}(\theta;\{x^i,e^i,y^i\})\right)$ approximates the generalization error. However, this formulation has a failure point. Consider a well-designed penalty term which successfully approximates the generalization gap. Clearly the joint optimization will decrease the generalization error. However, it can effectively increase the risk on seen domains due to the difficulties in optimization process. Moreover, if the reduction in generalization error is lower than the unintended excess risk on seen domains, the final performance on unseen domains can be worse than not doing any domain generalization. We hypothesize that this behavior might explain the recently demonstrated behavior from [23, 18].

As a motivating empirical finding, consider the benchmarking results from [23], which are enclosed in Appendix D. Take CORAL [42] as a representative method[1]. When we look at out-of-distribution performance, ERM outperforms CORAL on 7 out of 9 problems. However, when we look at the generalization gap (i.e., the difference between in-distribution and out-of-distribution performance), CORAL has lower gap than ERM on 7 out of 9 problems. Although the models learned with CORAL have lower generalization error, the loss of in-distribution performance is so significant that the resulting models do not perform well when tested out-of-distribution.

In order to partially address this issue, we propose a constraint of no excess risk. Instead of jointly minimizing the empirical risk and the penalty, we minimize the penalty under the constraint of the optimality of the empirical risk. More formally, we consider the following optimization problem:

$$\min_\theta \quad \texttt{Penalty}(\theta;\{x^i,e^i,y^i\})$$
$$st. \quad \theta \in \arg\min_\theta \mathcal{L}(\theta)$$

(3)

*Remarks on the reformulation:* i) This reformulation eases the aforementioned issue as it guarantees no excess risk over the seen domains. Since the optimality of the empirical risk is directly a constraint, this formulation would not harm in-distribution performance. ii) This reformulation is not sensible if performance on seen domains and unseen domains are inherently contradictory. For example, if there is strong nuisance information in the seen domains, any model not using this nuisance information will fail to optimize the empirical risk, effectively making excess risk a requirement for generalization. However, for the benchmarks we are interested in, which involve supervised learning from multiple datasets, this is not the case as a universal model solving all datasets (seen and unseen) exists. In other words, we address only a subset of domain generalization problems where seen and unseen domains do not compete. iii) Since the models we use are significantly overparameterized, the space of solutions to empirical risk minimization is quite large, likely including many domain-invariant and domain-sensitive solutions. Hence, this problem is still non-trivial from the numerical optimization perspective. iv) Finally, one can mistakenly think that our reformulation and the original joint optimization would find the same solution due to the Lagrangian function. However, this is incorrect as the Lagrangian function requires optimization of the Lagrangian multiplier, which is intractable for large-scale deep learning and typically replaced with fixed hyper-parameters. Moreover, we propose to directly solve (3) without resorting to Lagrangian function.

In the rest of this section, we first discuss in Section 2.3 how to convert the constraint in (3) into a practical form for first-order optimization. Then in Section 2.4 we discuss how (3) is closely related to rate-distortion theory, demonstrating an unexpected connection. Finally in Section 2.5 we utilize tools from rate-distortion theory to numerically solve (3).

---

[1]This choice is not binding: similar conclusions hold for other methods.

## 2.3   Stationarity of the ERM as a Constraint

The overarching objective of $\theta \in \arg\min_\theta \mathcal{L}(\theta)$ is rather too ambitious. Since the family of loss functions we are interested in is non-convex, the best we can hope is convergence to a stationary point. Hence, we convert (3) into

$$\min_\theta \quad \texttt{Penalty}(\theta; \{x^i, e^i, y^i\})$$
$$st. \quad \|\nabla_\theta \mathcal{L}(\theta)\| \le \epsilon \tag{4}$$

While solving the problem in (4), we constrain ourselves to iterative methods to be compatible with stochastic gradient descent (SGD), which is commonly used to train deep neural networks. Hence, we would like to have an iterative procedure $\theta^{t+1} = \theta^t - \eta G^t$ which solves (4) in convergence. Before we proceed, we note that when $G^t$ is the stochastic gradient (i.e., $\mathbb{E}[G^t] = \nabla_\theta \hat{\mathcal{L}}(\theta)$), convergence to a stationary point of the empirical risk can be shown under mild conditions. However, we are also interested in minimizing the penalty. Hence, we need to allow updates that are different than the gradients while still supporting eventual convergence.

Convergence of SGD with biased gradients has previously been analyzed by Sener & Koltun [37], Hu et al. [22], and Ajalloeian & Stich [1]. The common theme is the necessity for the bias to decrease over time. This requirement is explicit in [37] and implicit in [22] and [1]. In the implicit case, bias is assumed to be linearly bounded by the gradient norm, later shown to decrease.

We adapt the analysis of SGD with bias from Sener & Koltun [37] and show that as long as the norm of the bias is bounded and this bound decreases during training faster than the rate $1/\sqrt{t}$, it convergences to a stationary point of a smooth and non-convex risk function. We state the convergence result in the following proposition and defer the proof to Appendix A.

**Proposition 2.1.** *Apply stochastic updates $\theta^{t+1} = \theta^t - \eta G^t$ for $T$ steps. Assume i) $\mathcal{R}(\theta)$ is a non-convex, $L$-Lipschitz, and $\mu$-smooth function bounded by $\Delta$; ii) Updates are bounded in expectation, $\mathbb{E}[\|G^t\|_2^2] \le V$; iii) Bias of the updates is decreasing, $\|\mathbb{E}[G^t] - \nabla\mathcal{R}(\theta^t)\|_2 \le D/\sqrt{t}$. Then,*

$$\frac{1}{T}\sum_{t=1}^{T}\mathbb{E}[\|\nabla\mathcal{R}(\theta^t)\|_2^2] \le 2\left(\sqrt{\Delta\mu V} + LD\right)\sqrt{\frac{1}{T}}.$$

In this proposition, we assume that the risk function is non-convex, $L$-Lipschitz, and $\mu$-smooth[2], and that the stochastic updates are bounded. These are standard assumptions and shown to be valid for common deep learning architectural choices [19]. Finally, the third assumption is on the bias decreasing with the rate of $1/\sqrt{t}$, and we explicitly handle it in our solver, which we discuss below.

With Proposition 2.1, we operationalize the definition of a satisfactory iterative update. As long as we can bound the difference between the updates and the gradient of the empirical risk, we can guarantee convergence to a stationary point. Hence, our iterative approach to domain generalization starts with an initialization $\theta^0$ and applies the iteration $\theta^{t+1} = \theta^t - \eta G^t$, with $G^t$ being the solution of the following optimization problem:

$$\min_{p(G^t)} \quad \mathbb{E}_{G^t}\left[\texttt{Penalty}(\theta^t + G^t; \{x^i, e^i, y^i\})\right]$$
$$st. \quad \mathbb{E}_{G^t}\left[\|G^t - \nabla_\theta\mathcal{L}(\theta)\|_2\right] \le \frac{D}{\sqrt{t}} \tag{5}$$

The constraint of our final formulation in (5) guarantees the assumption (iii) from Proposition 2.1. Hence, applying the updates $G^t$, which are the solution of (5), would satisfactorily minimize the empirical risk. In the meantime, we seek the updates minimizing the penalty among these satisfactory ones. Since our approach utilizes satisfactory updates for empirical risk instead of the first-order optimal ones, following Herbert Simon's characterization of satisficing [40], we call this approach *Satisficing Domain Generalization* (SDG). Finally, the minimization is over the distribution of the updates instead of the update itself. Although this is unconventional, it is intentional and crucial for making the problem compatible with rate-distortion theory, which we utilize heavily in our solver.

---

[2]$L$-Lipschitz, $\mu$-smooth: $|\mathcal{R}(\theta^1) - \mathcal{R}(\theta^2)| \le L\|\theta^1 - \theta^2\|$, $\|\nabla\mathcal{R}(\theta^1) - \nabla\mathcal{R}(\theta^2)\| \le \mu\|\theta^1 - \theta^2\| \; \forall \theta^1, \theta^2$.

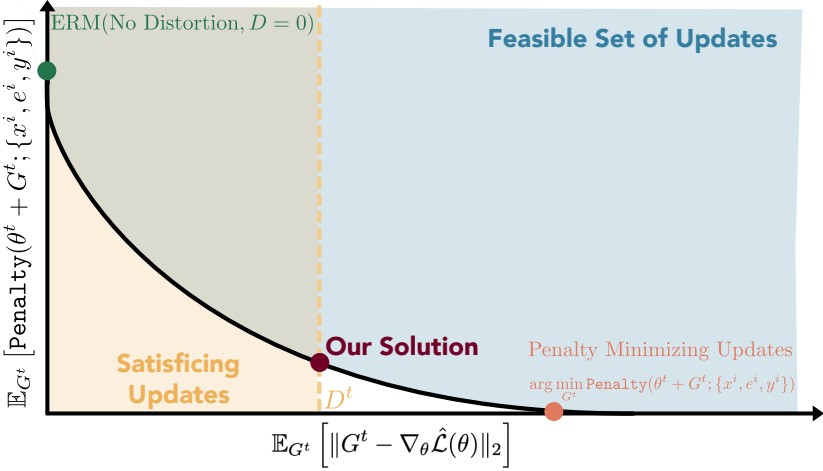

Figure 1: The penalty-distortion function $R(D)$ is a non-increasing and convex function of $D$ that describes the set of feasible updates. Our analysis of SGD with bias implies that the $\mathbb{E}[\|G^t - \nabla \mathcal{R}_e(\theta^t)\|_2] \leq D^t$ describes the set of updates that eventually minimize the empirical risk. The penalty minimizing updates can lie well beyond the region that solves ERM. Our formulation chooses the update that minimize the penalty while eventually solving the ERM problem.

## 2.4 Understanding Satisficing Domain Generalization

In this section, we provide observations on the proposed formulation (5) by utilizing tools from rate-distortion theory. Consider the function $R(D)$ as the solution of (5) for a given constant factor $D$. The problem (5) is simply minimizing the penalty with the constraint of the limited distortion in updates. This form is strikingly similar to the rate-distortion function, where the (compression) rate is optimized with the constraint of acceptable degradation in decoding. If the penalty were the mutual information between the true gradient and $G^t$, it would have been exactly equivalent. Hence, we can further utilize tools from rate-distortion theory to better understand (5). We now state a few simple facts derived from rate-distortion theory. (*Proofs follow their rate-distortion counterparts directly and are provided in Appendix B.*)

**Extreme cases:** When $D^t = D/\sqrt{t} = 0$, $E[G^t] = \nabla \mathcal{L}(\theta^t)$. Hence, at the end of training, we perform ERM updates. When the distortion is unbounded ($D^t = \infty$), updates directly minimize the penalty. Moreover, our proposed formulation smoothly goes from updates minimizing penalty to ERM updates during training.

**Improvement guarantee:** $R(D)$ is a non-increasing and convex function of $D$. Hence, $R(0) \geq R(D^t)$. In other words, SDG decreases the penalty when compared with ERM.

**Geometric picture:** The satisficing requirement implies a certain structure over the plane where the $x$-axis is the distortion and the $y$-axis is the penalty. We visualize this geometry and the interpretation of our method in Figure 1.

## 2.5 Solving Satisficing Domain Generalization

We present a numerical algorithm to solve (5). We first give an iterative method similar to Blahut-Arimoto (BA) [11, 4] with intractable complexity. We later make simplifying assumptions to design a tractable method.

We propose to regularize the objective with the mutual information between gradients ($G^t$) and the domain-ids ($E$). This regularization is sensible as we would like gradients to carry little information about the domain-ids. The final numerical optimization problem we tackle is

$$\min_{p(G_i^t)} \quad \mathbb{E}_{G^t}\left[\texttt{Penalty}(\theta^t + G^t; \{x^i, e^i, y^i\})\right] + \gamma \mathcal{I}(G^t; E)$$

$$st. \quad \mathbb{E}_{G^t}\left[\|G^t - \nabla_\theta \mathcal{L}(\theta)\|_2\right] \leq \frac{D}{\sqrt{t}}$$

(6)

The information regularization is not only desired but also crucial as it enables us to apply technical derivation similar to Blahut-Arimoto [11, 4].

To solve (6), We only consider the discrete and finite probability mass functions where $G^t \in \{G_1^t, \ldots, G_K^t\}$; hence, we only need to solve for the point masses $p_{G_i^t} = p(G^t = G_i^t)$. We first define auxiliary variables $p_{G_i^t|e} = p(G^t = G_i^t|E = e)$, which are conditioned on domain-ids. By this decomposition, we can solve for each domain conditional probability $p_{G_i^t|e}$ separately and later marginalize them via $p_{G_i^t} = {}^1/_M \sum_e p_{G_i^t|e}$. We further replace $\mathbb{E}_{G^t}[\|G^t - \nabla_\theta \hat{\mathcal{L}}(\theta)\|_2] \leq {}^D/\sqrt{t}$ with a stronger constraint of uniform bounds over domains as $\mathbb{E}_{G^t|E=e}[\|G^t - \nabla_\theta \hat{\mathcal{L}}^e(\theta)\|_2] \leq {}^D/\sqrt{t}$ and apply a similar derivation as Blahut [11] and Arimoto [4]. We defer the complete derivation to the Appendix C.1 and state the update steps directly.

Consider the notation $d(G, e) = \|G - \nabla \mathcal{L}^e(\theta)\|_2$, and $\texttt{Pen}(G) = \texttt{Penalty}(\theta^t + G; \{x^i, e^i, y^i\})$. We initialize the conditional probabilities uniformly as $p_{G_i^t|e} = {}^1/_K$ and apply the iterations:

- Domain Specific Gradients: $\hat{p}_{G_k^t|e}^{l+1} = p_{G_k^t}^l \exp\left[-\frac{1}{\gamma}\left(\texttt{Pen}(G_k^t) + \beta d(G_k^t, e)\right)\right]$

- Normalization: $p_{G_k^t|e}^{l+1} = \dfrac{\hat{p}_{G_k^t|e}^{l+1}}{\sum_{\hat{k}} \hat{p}_{G_{\hat{k}}^t|e}^{l+1}}$

- Marginalization: $p_{G_k^t}^{l+1} = {}^1/_M \sum_e p_{G_k^t|e}^{l+1}$

Here $\beta^t$ is a parameter that inversely depends on $D^t$. These iterations converge to a solution where all the conditional distributions are equal.

Applying these updates to the space of gradients is intractable since the space of $G^t$ is high-dimensional. We propose two simplifications to make the application of BA tractable: i) we solve for each parameter independently, and ii) we only estimate the sign and copy the magnitude from the ERM. Specifically, we solve the BA problem using the discrete set $(G^t)_p \in \{(G_1^t)_p = (\nabla \mathcal{L})_p, (G_2^t)_p = -(\nabla \mathcal{L})_p\}$ for $(G^t)_p$, which is the update for the $p^{th}$ parameter. We use uniform initialization: $p^0((G_1^t)_p) = p^0((G_2^t)_p) = 0.5$. We present a more detailed discussion in Appendix C.1&C.2 with complete derivation and give pseudocode in Algorithm 1. After the BA iterations are completed, we sample the update $G^t \sim p(G^t)$ and apply it to the model.

To understand the proposed simplifications, we consider their implications for the rest of the pipeline. We feed the resulting estimated gradients to first-order numerical optimizers. Moreover, we utilize the first-order approximation of the penalty function in our algorithms in Appendix C.1&C.2. Hence, the rest of the pipeline is first order, not utilizing higher-order relationships between coordinates, justifying the first simplification. The second simplification is similar to SignSGD [10] like methods, where only the sign of the gradient is used without its magnitude for efficient communication between worker nodes in multi-node optimization. Sign of the gradient suffices for convergence in non-convex problems both theoretically and empirically.

## 3 Experimental Results

### 3.1 Implementation details.

We implement the proposed algorithm in PyTorch. Following the analysis in Proposition 2.1, $D^t = D\sqrt{{}^1/_t}$. Moreover, $\beta$ is inversely related to $D$. We choose $\beta^t = (\beta^\circ/\sqrt{T})\sqrt{t}$ for epoch $t$ from $t = 1$ to $t = T$. We treat $\beta^\circ$ as a hyperparameter and search in $\beta^\circ \in \{0.01, 0.1, 1.0\}$ together with all other hyperparameters. Instead of treating $\gamma$ (weight of the information regularization) as a hyper-parameter, we present a simple heuristic to set it. Specifically, since the relevant term in the iterative update is $\exp\left[-\frac{1}{\gamma}\left(\texttt{InvP}(G_k^t) + \beta d(G_k^t, e)\right)\right]$, we keep track of the average of $\left(\texttt{InvP}(G_k^t) + \beta d(G_k^t, e)\right)$ throughout training and set it as $\gamma$ at each instant. In all of our experiments, we used 25 BA iterations. Since using all domains at each batch is not feasible, we first sample $B_D$ domains, then sample $B$ examples per domain (named as group sampler in WILDS[23]). We search all hyperparameters with random sampling with the constraint that all methods have the same budget for the hyperparameter search. Specifically, we use 20 random hyperparameter choices. We also

Table 1: **Out-of-distribution test performance in the WILDS benchmarks** [23]. We report average over 3 seeds for OGB-MolPCBA, FMoW, Amazon, and py150, 5 folds for PovertyMap, and 10 seeds for Camelyon17. We report standard deviations as uncertainty.

| | iWildCam | Camelyon17 | OgbMolPcba | FMoW | Amazon | Py150 | PovertyMap |
|---|---|---|---|---|---|---|---|
| | Macro-F1 | Avg Acc | AP | WorstR Acc | $10^{th}$-tile Acc | MethodClass | WstPearsonR |
| CORAL | $32.8 \pm 0.1$ | $59.5 \pm 7.7$ | $17.9 \pm 0.5$ | $31.0 \pm 0.4$ | $52.9 \pm 0.8$ | $65.9 \pm 0.1$ | $0.44 \pm 0.06$ |
| +AND | $8.2 \pm 0.5$ | $70.8 \pm 6.7$ | $10.3 \pm 0.1$ | $28.0 \pm 1.0$ | $51.5 \pm 0.7$ | $66.7 \pm 0.6$ | $0.44 \pm 0.03$ |
| +SAND | $\mathbf{33.9 \pm 1.4}$ | $64.7 \pm 4.2$ | $23.4 \pm 0.7$ | $\mathbf{33.3 \pm 0.2}$ | $53.1 \pm 0.5$ | $66.9 \pm 0.2$ | $0.44 \pm 0.08$ |
| +SDG | $\mathbf{34.6 \pm 1.2}$ | $\mathbf{75.2 \pm 4.4}$ | $25.8 \pm 0.1$ | $\mathbf{33.9 \pm 0.9}$ | $\mathbf{53.7 \pm 0.6}$ | $\mathbf{68.3 \pm 0.1}$ | $\mathbf{0.49 \pm 0.05}$ |
| FISH | $22.0 \pm 1.8$ | $\mathbf{74.7 \pm 7.1}$ | $22.6 \pm 0.3$ | $\mathbf{34.6 \pm 0.2}$ | $53.3 \pm 0.8$ | $65.6 \pm 0.1$ | $0.35 \pm 0.03$ |
| +SDG | $32.9 \pm 0.2$ | $\mathbf{76.2 \pm 8.4}$ | $26.1 \pm 0.5$ | $\mathbf{34.6 \pm 0.3}$ | $53.5 \pm 0.6$ | $66.5 \pm 0.2$ | $0.46 \pm 0.03$ |
| ERM | $31.0 \pm 1.3$ | $70.3 \pm 6.4$ | $\mathbf{27.2 \pm 0.3}$ | $32.8 \pm 0.5$ | $\mathbf{53.8 \pm 0.8}$ | $67.9 \pm 0.1$ | $0.45 \pm 0.06$ |

Table 2: **In-distribution test performance in the WILDS benchmarks** [23]. We report average over 3 seeds for OGB-MolPCBA, FMoW, Amazon, and py150, 5 folds for PovertyMap, and 10 seeds for Camelyon17. We report standard deviations as uncertainty.

| | iWildCam | Camelyon17 | OgbMolPcba | FMoW | Amazon | Py150 | PovertyMap |
|---|---|---|---|---|---|---|---|
| | Macro-F1 | Avg Acc | AP | WorstR Acc | $10^{th}$-tile Acc | MethodClass | WstPearsonR |
| CORAL | $43.5 \pm 3.5$ | $95.4 \pm 3.6$ | $18.4 \pm 0.2$ | $55.0 \pm 1.0$ | $55.1 \pm 0.4$ | $70.6 \pm 0.1$ | $0.59 \pm 0.03$ |
| +AND | $11.1 \pm 0.7$ | $96.6 \pm 0.1$ | $10.6 \pm 0.1$ | $48.9 \pm 0.2$ | $53.8 \pm 0.5$ | $73.4 \pm 0.2$ | $0.57 \pm 0.01$ |
| +SAND | $47.8 \pm 0.7$ | $\mathbf{97.9 \pm 0.2}$ | $28.0 \pm 0.2$ | $55.4 \pm 0.2$ | $56.3 \pm 0.5$ | $74.2 \pm 0.9$ | $0.59 \pm 0.02$ |
| +SDG | $\mathbf{48.1 \pm 1.0}$ | $95.5 \pm 3.8$ | $\mathbf{28.3 \pm 0.1}$ | $56.0 \pm 3.1$ | $56.6 \pm 0.5$ | $\mathbf{75.4 \pm 0.1}$ | $\mathbf{0.60 \pm 0.02}$ |
| ERM | $47.0 \pm 1.4$ | $93.2 \pm 5.2$ | $27.8 \pm 0.1$ | $\mathbf{58.3 \pm 0.9}$ | $\mathbf{57.3 \pm 0.1}$ | $\mathbf{75.4 \pm 0.4}$ | $0.57 \pm 0.07$ |

perform early stopping and choose the best epoch using validation domain performance. Moreover, to ensure a fair comparison, all methods are run for the same amount of wall-clock time. Since our method is slower, we perform fewer epochs than other methods. We list all chosen hyperparameters in Appendix F.

## 3.2 Evaluation on WILDS

**Datasets** Our main evaluation is using the WILDS benchmark [23]. WILDS is designed to evaluate domain generalization and subpopulation shift in realistic problems. Among these problems, we use seven problems that are either pure domain generalization problems or a combination of domain generalization and subpopulation shift. These problems cover a wide range, including outdoor images (iWildCam), medical images (Camelyon), satellite images (FMoW, PovertyMap), natural language (Amazon), formal language (py150), and graph-structured data (OGB-MolPCBA) with the number of domains from 5 to $120K$. The summary of the benchmarks is in Appendix E.

**Baselines** We apply our proposed optimizer (SDG) to CORAL [42] and denote CORAL+SDG. We compare CORAL+SDG with i) CORAL: the standard implementation of CORAL which is jointly minimizing the loss and the penalty, ii) CORAL+AND-MASK: Applying the AND-MASK[30] to CORAL minimization, iii) CORAL+SAND-MASK: Applying the SAND-MASK[38] to CORAL minimization. It is important to note that AND-MASK[30] and SAND-MASK[38] have not been applied in conjunction with CORAL before. Still, we compare with them since they also directly modify the optimization procedure. We compute the gradient of the empirical risk plus coral penalty for each domain separately and utilize their *and-mask* and *sand-mask* as corresponding updates.

### 3.2.1 Main Results

We summarize the out-of-distribution results in Table 1 and in-distribution results in Table 2. We also compute the generalization gap (difference between in-distribution and out-of-distribution performance) and tabulate it in Table 3. When evaluated on out-of-distribution performance, our method improves CORAL in all settings. The improvement is significant for Camelyon17 with more than 15% and OGB-MolPCBA with more than 7%. Moreover, although the CORAL outperforms ERM with a standard optimizer only for 1 out 7 benchmarks, combined with our optimizer, CORAL outperforms ERM for 6 out of 7 benchmarks. Our method closes the gap for Camelyon17, FMoW,

Amazon, Py150, and PovertyMap. The only setting CORAL fails to outperform ERM when combined with our optimizer is OGBMolPCBA. We believe one reason for this is the additional computational complexity of our method. For a fair comparison, our method performs significantly fewer epochs than ERM. When compared with AND-mask and SAND-mask, our method outperforms them in all benchmarks.

When evaluated for in-distribution performance in Table 2, our method improves upon CORAL in all benchmarks. This is rather expected as our approach explicitly ensures the optimality of empirical risk. When the generalization gap (difference between in-distribution and out-of-distribution) is evaluated in Table 3, our method has a similar generalization gap with CORAL in average (our method has a smaller gap for 3 benchmarks, similar gap within the stan-

Table 3: Generalization Gap (difference between in-distribution and out-distribution performance) in the WILDS benchmarks [23]. We denote the results which generalize better than ERM with Green color and worse than ERM with Red color.

|        | iWildC. | Cam  | Mol | FMoW | Amazon | Py150 | Pov.Map |
|--------|---------|------|-----|------|--------|-------|---------|
| CORAL  | 10.7    | 35.9 | 0.5 | 24   | 2.2    | 4.7   | 0.15    |
| +AND   | 2.9     | 25.8 | 0.3 | 20.9 | 2.3    | 6.7   | 0.13    |
| +SAND  | 13.9    | 33.2 | 4.6 | 22.1 | 3.2    | 7.3   | 0.15    |
| +SDG   | 13.5    | 20.3 | 2.5 | 22.1 | 2.9    | 7.1   | 0.11    |
| ERM    | 16      | 22.9 | 0.6 | 25.5 | 3.5    | 7.5   | 0.12    |

dard deviation for 2, and a larger gap for other 2). Hence, our proposed optimizer improves empirical risk without hurting the generalization ability on 5 out of 7 datasets. Even when our method hurts generalization, the improvement on the empirical risk is so large the resulting out-of-distribution performance improves.

### 3.3 Additional Empirical Analysis

In addition to our main study of applying our method to CORAL in WILDS benchmark, here we test the generality of our method for penalty function choice (i.e. generality beyond CORAL) as well as application (i.e. generality beyond WILDS).

**Applicability beyond CORAL** We test generality of our method beyond CORAL. Specifically, we apply our method to FISH [39] model in WILDS benchmark [23] and report in Table 1. Application to FISH [39] requires modifications as it is not directly penalty based and we explain them in Appendix C.3. Results on FISH [39] suggest that our method improves FISH on 5 out of 7 benchmarks. Moreover, the improvement for iWildCam and PovertyMap is significant, more than 9%.

**Applicability beyond WILDS** We test generality of our method beyond WILDS benchmark. Specifically, we apply our method to FISH [39], VRex [25], and CORAL [42] model in DomainBed [18] benchmark. We directly use the benchmark code shared by DomainBed [18] and follow the official process. We tabulate the results in Table 4. The results suggest that our method consistently improves the performance of all tested methods.

Consistent improvement for 3 different domain generalization models on 14 different benchmarks over 2 experimental suites conclusively demonstrates the generality of the proposed method.

Table 4: **Out-of-distribution test performance of VRex** [25] **in the DomainBed benchmarks** [18]. We average over 3 seeds and report standard deviations as uncertainty.

| Algorithm | C-MNIST | R-MNIST | VLCS | PACS | OfficeHome | TerraIncog. | DomainNet | Avg |
|-----------|---------|---------|------|------|-----------|-------------|-----------|-----|
| FISH      | $51.6 \pm 0.1$ | $98.0 \pm 0.0$ | $77.8 \pm 0.3$ | $85.5 \pm 0.3$ | $68.6 \pm 0.4$ | $45.1 \pm 1.3$ | $42.7 \pm 0.2$ | 67.1 |
| +SDG      | $51.9 \pm 0.1$ | $98.0 \pm 0.0$ | $\mathbf{79.9 \pm 0.5}$ | $\mathbf{87.3 \pm 0.3}$ | $68.4 \pm 0.5$ | $48.4 \pm 0.8$ | $42.6 \pm 0.4$ | 68.1 |
| CORAL     | $51.5 \pm 0.1$ | $98.0 \pm 0.1$ | $78.8 \pm 0.6$ | $86.2 \pm 0.3$ | $\mathbf{68.7 \pm 0.3}$ | $47.6 \pm 1.0$ | $41.5 \pm 0.1$ | 67.5 |
| +SDG      | $52.5 \pm 0.2$ | $\mathbf{98.2 \pm 0.1}$ | $79.7 \pm 0.3$ | $86.9 \pm 0.4$ | $\mathbf{68.7 \pm 0.7}$ | $49.9 \pm 1.2$ | $41.4 \pm 0.4$ | $\mathbf{68.2}$ |
| VREx      | $51.8 \pm 0.1$ | $97.9 \pm 0.1$ | $78.3 \pm 0.2$ | $84.9 \pm 0.6$ | $66.4 \pm 0.6$ | $46.4 \pm 0.6$ | $33.6 \pm 2.9$ | 65.6 |
| +SDG      | $\mathbf{52.7 \pm 0.1}$ | $98.1 \pm 0.2$ | $78.9 \pm 0.5$ | $86.4 \pm 0.3$ | $68.2 \pm 0.5$ | $48.1 \pm 0.8$ | $\mathbf{41.8 \pm 0.4}$ | 67.7 |
| ERM       | $51.5 \pm 0.1$ | $98.0 \pm 0.0$ | $77.5 \pm 0.4$ | $85.5 \pm 0.2$ | $66.5 \pm 0.3$ | $46.1 \pm 1.8$ | $40.9 \pm 0.1$ | 66.6 |

### 3.3.1 Impact on training time.

Although our method effectively learns generalizable models, it has a rather significant impact on computational complexity. Consider an input batch of $B \times B_D$ datapoints over $B_D$ domains. Direct optimization only needs the gradient for the average risk, while our method requires a gradient for each domain. Hence, our method needs $B_D$ backward passes, whereas direct optimization needs only a single pass. Effectively, our training time is up to $B_D$ times slower. It is important to note that our method has very limited impact on training time when the original algorithm computes gradient per domain. For example, utilizing our method with IRM would result in small change in training time.

To empirically evaluate the computational impact, we compute the average time per epoch per dataset and report it for our method and ERM in Table 5. Our method increases the training time significantly. Although this

Table 5: Wall-clock time (in minutes) for training a single epoch for CORAL and CORAL+SDG method on an RTX 3090 GPU. Since our method needs gradients per-domain, it needs number of domains in the batch times backward pass during training. This is a limitation of our method.

|  | CORAL | CORAL+SDG |
|---|---|---|
| iWildCam | 57 | 91 |
| Camelyon17 | 19 | 72 |
| OGB-MolPCBA | 22 | 73 |
| FMoW | 9 | 41 |
| Amazon | 100 | 227 |
| Py150 | 186 | 420 |
| PovertyMap | 4 | 9 |

limitation can be easied using some approximation techniques [36], we leave the potential computational improvements as future work. Moreover, it is important to clarify that we decrease the number of epochs for our method to ensure all numbers are fair in terms of used compute.

## 4 Related Work

**Domain generalization.** Domain generalization addresses settings where the training and test data come from different distributions, violating the common independent and identically distributed (iid.) assumption. The training data comes from multiple distinct data distributions (called domains). The typical assumption to tackle domain generalization is covariate shift [9], where only the data distribution changes between domains and the conditional distribution of label given data stays the same. We are specifically interested in penalty/invariance based domain domain generalization. However, the approaches for domain generalization are more general and include domain conditioning [28], self-supervised pre-training [3], domain-specific decomposition [32], and data augmentation [44]. We treat these works as orthogonal to us. One approach for invariance based domain generalization is *distributionally robust optimization*. Here a robust form of risk is optimized instead of the average risk [35, 45, 16]. The robustness is typically against a predefined amount of distributional shift around the training data, hoping that it covers the test distribution. Similarly, Krueger et al. [25] enforce uniformity of risk functions to guarantee robustness following a causality-inspired approach. Another common approach is *adversarial feature learning*. Here a representation learning problem is posed such that an adversary cannot detect the domain information from the learned features. Adversarial feature learning is achieved via game-theoretic min-max formulations [17, 26], distributional distances such as MMD [2], and divergence of second-order statistics of features [42]. *Causality* inspired methods is another thrust where the problem is posed in terms of finding features following the causal direction. The out-of-distribution generalization problem considered in causality-based methods addresses a larger class of problems beyond domain generalization since they are applicable beyond covariate shift. An early application of causality to out-of-distribution generalization was for linear models [31], later extended to the non-linear case [5]. Final category which we discuss is *gradient alignment*. Here gradient-based learning is directly addressed and algorithms to align gradients of risk functions from different domains are developed. This class of methods is the most similar to our case. Koyama & Yamaguchi [24] enforce gradient similarity by minimizing the variance of the gradients and Shi et al. [39] minimize the cosine distance between gradients. Similarly, [50] finds domain transferrable solutions using meta-learning. Parascandolo et al. [30] combine these ideas with causality-based motivation and propose a method that directly manipulates gradients. Among these methods, ANDMask [30] and SANDMask [38] are the most relevant to us since they are the only ones that explicitly modify gradients and use sign information.

**Rate-distortion approaches.** Rate-distortion theory addresses the problem of lossy compression within information theory [13]. Among approaches that are numerical and directly applicable to any distribution, the Blahut-Arimoto algorithm is a common choice [11, 4]. Although there are

improved variants [46, 48, 27], we find the original algorithm simple and effective. To the best of our knowledge, our work is the first application of the Blahut-Arimoto algorithm to the problem of domain generalization. The only application of Blahut-Arimoto within machine learning that we are aware of is the BLAST algorithm [6], which uses the information-theoretic analysis of Thompson sampling [34] for better sample efficiency. Ideas from rate-distortion theory have recently been applied to the problem of learning invariant representations by Zhao et al. [51] to understand the underlying tradeoffs of transfer learning theoretically. Different from [51], we analyze convergence properties in addition to the information-theoretic motivation. We also propose a practical method for transfer learning. A closely related problem to rate-distortion is the information bottleneck [43], where the mutual information between representation and input data is minimized while keeping the mutual information between representation and label maximal. Although these methods have partially been applied to domain adaptation and generalization setting [14, 41, 15, 52], they typically require stochastic neural networks and additional modeling assumptions; hence, they are not directly applicable to our case. To the best of our knowledge, these methods have not yet been applied to large-scale problems on the scale that we are interested in. Our approach is significantly different from these approaches as our method stems from an analysis of gradient descent and an explicit formulation of satisficing gradient descent.

## 5   Conclusion

We considered penalty-based domain generalization and reformulated it to find updates minimizing the surrogate penalty under the constraint that applying these updates eventually converges to the stationary point of the empirical risk. We developed an optimizer that is applicable to existing domain generalization models. Our extensive empirical analysis on the WILDS and DomainBed benchmark validates the efficacy of our method for CORAL, FISH and VRex models.

A major drawback of our method is the additional computational complexity as it needs gradients per domain. A possible future direction is applying approximation techniques to reduce this complexity. Moreover, we provided an exciting connection between rate-distortion theory and domain generalization. Another interesting future direction is utilizing ideas from the rate-distortion theory literature to better understand and improve domain generalization.

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
