# A Proof of Proposition 2.1

Consider applying the stochastic updates $\theta^{t+1} = \theta^t - \eta G^t$ to a function $\mathcal{R}(\theta)$. Using the $\mu$-smoothness of the function $\mathcal{R}$,

$$\mathcal{R}(\theta^{t+1}) = \mathcal{R}(\theta^t - \eta G^t) \leq \mathcal{R}(\theta^t) - \eta \nabla \mathcal{R}(\theta^t)^\mathsf{T} G^t + \frac{\mu \eta^2}{2} \|G^t\|_2^2. \tag{7}$$

Taking the expectation of the inequality and using the fact that $\mathbb{E}[\|G^t\|_2^2] \leq V$,

$$\mathbb{E}[\mathcal{R}(\theta^{t+1})] \leq \mathbb{E}[\mathcal{R}(\theta^t)] - \eta \mathbb{E}[\|\nabla \mathcal{R}(\theta^t)\|_2^2] - \eta \mathbb{E}[\nabla \mathcal{R}(\theta^t)^\mathsf{T} (G^t - \nabla \mathcal{R}(\theta^t))] + \frac{\mu \eta^2 V}{2}. \tag{8}$$

Using the Lipschitz smoothness of the $\mathcal{R}$ and re-ordering the terms,

$$\eta \mathbb{E}[\|\nabla \mathcal{R}(\theta^t)\|_2^2] \leq \mathbb{E}[\mathcal{R}(\theta^t)] - \mathbb{E}[\mathcal{R}(\theta^{t+1})] + \frac{\mu \eta^2 V}{2} + \eta L \left\| \mathbb{E}[G^t] - \nabla \mathcal{R}(\theta^t) \right\|_2. \tag{9}$$

Summing up from $t = 1$ to $T$ and dividing by $\eta$, we obtain

$$\sum_{t=1}^T \mathbb{E}[\|\nabla \mathcal{R}(\theta^t)\|_2^2] \leq \frac{2\Delta}{\eta} + \frac{\mu \eta T V}{2} + L \sum_{t=1}^T \|\mathbb{E}[G^t] - \nabla \mathcal{R}(\theta^t)\|_2. \tag{10}$$

Using the fact that the distortions are bounded as $D^t = \|\mathbb{E}[G^t] - \nabla \mathcal{R}(\theta^t)\|_2 \leq D/\sqrt{t}$, we can bound the sum $\sum_{t=1}^T D^t$ with $2D\sqrt{T}$. By solving for the optimal $\eta$ as;

$$\eta = 2\sqrt{\frac{\Delta}{\mu T V}}. \tag{11}$$

The final bound becomes,

$$\sum_{t=1}^T \mathbb{E}[\|\nabla \mathcal{R}(\theta^t)\|_2^2] \leq \frac{2\Delta}{\eta} + \frac{\mu \eta T V}{2} + L \sum_{t=1}^T \|\mathbb{E}[G^t] - \nabla \mathcal{R}(\theta^t)\|_2. \tag{12}$$

We plug the resulting step size in the bound and divide it with $T$, to obtain,

$$\frac{1}{T} \sum_{t=1}^T \mathbb{E}[\|\nabla \mathcal{R}(\theta^t)\|_2^2] \leq 2 \left( \sqrt{\Delta \mu V} + LD \right) \sqrt{\frac{1}{T}} \tag{13}$$

# B Proofs of Statements from Section 2.4

In this section, we prove the facts that we state in Section 2.4 without proof. These proofs straightforwardly follow their rate-distortion version from [13].

We are interested in understanding the following function;

$$R(D) \triangleq \min_{p(G^t)} \quad \mathbb{E}_{G^t} \left[ \texttt{Penalty}(\theta^t + G^t; \{x^i, e^i, y^i\}) \right]$$

$$\text{st.} \quad \mathbb{E}_{G^t} \left[ \|G^t - \nabla_\theta \hat{\mathcal{L}}(\theta)\|_2 \right] \leq \frac{D}{\sqrt{t}} \tag{14}$$

We state the formal versions of the informal statements in Section 2.4 and their proofs as follows;

**Proposition B.1** (Extreme Cases)**.** *i) When $D^t = D/\sqrt{t} = 0$, $E[G^t] = \nabla \mathcal{L}(\theta^t)$. ii) When the distortion is unbounded ($D^t = D = \infty$), updates follow a distribution with the minimum penalty in expectation.*

*Proof for i.* When $D^t = D = 0$, the constraint in (14) becomes,

$$\mathbb{E}_{G^t} \left[ \|G^t - \nabla_\theta \hat{\mathcal{L}}(\theta)\|_2 \right] \leq 0 \tag{15}$$

The norm is nonnegative, and a nonnegative random variable has zero expectation if and only if it is zero almost surely. Hence,

$$p(G^t = \nabla_\theta \hat{\mathcal{L}}(\theta)) = 1 \tag{16}$$

$\square$

*Proof for ii.* When $D^t = D = \infty$, the constraint in (14) is trivially satisfied for all distribution as we assume $G^t$ is finite. Hence, (14) is equivalent to;

$$R(D) = \min_{p(G^t)} \mathbb{E}_{G^t} \left[ \texttt{Penalty}(\theta^t + G^t; \{x^i, e^i, y^i\}) \right] \tag{17}$$

where the solution is a distribution with the minimum penalty in expectation. $\square$

**Proposition B.2** (Monotonicity and Convexity). *$R(D)$ is a non-increasing and convex function of $D$.*

*Proof.* $R(D)$ is minimum of the penalty over increasingly larger sets as $D$ increases. Hence, $R(D)$ is non-increasing in $D$.

In order to prove convexity, consider two solutions $p(G_1^t)$ and $p(G_2^t)$, minimizing $R(D)$ for given $D_1^t, D_2^t$. Now, consider the distribution $p(G_\lambda^t) = \lambda p(G_1^t) + (1 - \lambda)p(G_2^t)$ and $D_\lambda^t = \lambda D_1^t + (1 - \lambda)D_2^t$. Since the $R(D_\lambda^t)$ is minimization over all distributions, it is smaller than penalty for $p(G_\lambda^t)$. In other words,

$$R(D_\lambda^t) \leq \mathbb{E}_{G^t \sim p(G_\lambda^t)} \left[ \texttt{Penalty}(\theta^t + G^t; \{x^i, e^i, y^i\}) \right] \tag{18}$$

Since expectation is a linear operator,

$$\mathbb{E}_{G^t \sim p(G_\lambda^t)} \left[ \texttt{Penalty}(\theta^t + G^t; \{x^i, e^i, y^i\}) \right]$$
$$= \lambda \mathbb{E}_{G^t \sim p(G_1^t)} \left[ \texttt{Penalty}(\theta^t + G^t; \{x^i, e^i, y^i\}) \right] + (1 - \lambda) \mathbb{E}_{G^t \sim p(G_2^t)} \left[ \texttt{Penalty}(\theta^t + G^t; \{x^i, e^i, y^i\}) \right] \tag{19}$$

Since $p(G_1^t)$ and $p(G_2^t)$ are the solutions for the corresponding minimization problems,

$$\mathbb{E}_{G^t \sim p(G_1^t)} \left[ \texttt{Penalty}(\theta^t + G^t; \{x^i, e^i, y^i\}) \right] = R(D_1^t)$$
$$\mathbb{E}_{G^t \sim p(G_2^t)} \left[ \texttt{Penalty}(\theta^t + G^t; \{x^i, e^i, y^i\}) \right] = R(D_2^t) \tag{20}$$

Combining these two statements proves the convexity as;

$$R(\lambda D_1^t + (1 - \lambda)D_2^t) \leq \lambda R(D_1^t) + (1 - \lambda)R(D_2^t) \tag{21}$$

$\square$

# C   Missing Derivations for Blahut-Arimoto and its Application to CORAL, FISH and VRex.

## C.1   Deriving Blahut-Arimoto Style Method to Solve (6)

We develop a Blahut-Arimoto [11, 4] style method to solve the following problem:

$$\min_{p(G^t)} \quad \mathbb{E}_{G^t} \left[ \texttt{Penalty}(\theta^t + G^t; \{x^i, e^i, y^i\}) \right] + \gamma \mathcal{I}(G^t; E)$$
$$st. \quad \mathbb{E}_{G^t} \left[ \|G^t - \nabla_\theta \hat{\mathcal{L}}(\theta)\|_2 \right] \leq \frac{D}{\sqrt{t}} \tag{22}$$

Before we apply the Blahut-Arimoto technique, we first utilize the definition of mutual information to convert (22) into

$$\min_{p(G^t)} \quad \mathbb{E}_{G^t}\left[\texttt{Penalty}(\theta^t + G^t; \{x^i, e^i, y^i\})\right] + \gamma \mathbb{E}_E[\mathcal{D}_{KL}(p(G|E)||p(G))]$$

$$st. \quad \mathbb{E}_{G^t}\left[\|G^t - \nabla_\theta \hat{\mathcal{L}}(\theta)\|_2\right] \leq \frac{D}{\sqrt{t}} \tag{23}$$

where $\mathcal{D}_{KL}$ is the KL-Divergence.

The key technique used in the Blahut-Arimoto method is decomposing $p(G^t)$ as $p(G^t) = \sum_e p(G^t|E = e)p(E = e)$. Denote $d(G^t, e) = \|G^t - \nabla \mathcal{L}^e(\theta)\|_2$ where $\mathcal{L}^e$ is the loss for domain $e$ and $|E|$ as the number of domains. Moreover, we assume all domains are equal in importance as $p(E = e) = {}^1/|E|$. Using the fact that the problem is discrete, $G^t \in \{G_1, \dots, G_K\}$, the problem in (22) can be transformed into

$$\min_{p(G_k|E=e)} \sum_e \sum_{k=1}^{K} p(G_k|E = e)\texttt{Penalty}(\theta^t + G_k; \{x^i, e^i, y^i\}) + \gamma \sum_e \sum_{k=1}^{K} p(G_k|E = e) \log \frac{p(G_k|E = e)}{p(G_k)}$$

$$st. \quad \sum_e \sum_{k=1}^{K} p(G_k|E = e)d(G_k, e) \leq \frac{D|E|}{\sqrt{t}} \tag{24}$$

here we replaced $\mathbb{E}_{G^t}\left[\|G^t - \nabla_\theta \hat{\mathcal{L}}(\theta)\|_2\right] \leq \frac{D}{\sqrt{t}}$ with a stronger constraint of uniform bounds over domains as $\mathbb{E}_{G^t|E=e}\left[\|G^t - \nabla_\theta \hat{\mathcal{L}}^e(\theta)\|_2\right] \leq \frac{D}{\sqrt{t}}$. We further compute the Lagrangian of this optimization problem with the Lagrange multiplier $\beta$ as;

$$\min_{p(G_k|E=e)} \max_{\beta \geq 0} \sum_e \sum_{k=1}^{K} p(G_k|E = e)\left[\texttt{Penalty}(\theta^t + G_k; \{x^i, e^i, y^i\}) + \dots\right.$$

$$\left.\dots + \gamma \frac{\log p(G_k|E = e)}{p(G_k)} + \beta d(G_k, e)\right] - \beta \frac{D|E|}{\sqrt{t}} \tag{25}$$

We take the derivative with respect to $p(G_k|E = e)$ and equate it to 0. Ignoring the constant factor,

$$p(G_k|E = e) \sim p(G_k) \exp\left[-\frac{1}{\gamma}\left(\texttt{Penalty}(\theta^t + G_k; \{x^i, e^i, y^i\}) + \beta d(G_k, e)\right)\right] \tag{26}$$

Since we know the probabilities sum to 1, we can handle the normalization factor separately. Start with initialization $p^\circ(G_k|E = e)$, which is typically uniform unless a prior information exists. Then, the following iterations sove (22),

- $\hat{p}^{l+1}(G_k|E = e) = p^l(G_k) \exp\left[-\frac{1}{\gamma}\left(\texttt{Penalty}(\theta^t + G_k; \{x^i, e^i, y^i\}) + \beta d(G_k, e)\right)\right]$

- $p^{l+1}(G_k|E = e) = \frac{\hat{p}^{l+1}(G_k|E=e)}{\sum_{\hat{k}} \hat{p}^{l+1}(G_{\hat{k}}|E=e)}$

- $p^{l+1}(G_k) = {}^1/|E| \sum_{E=e} p^{l+1}(G_k|E = e)$

## C.2 Applying Blahut-Arimoto Style Method to Penalty Based Deep Domain Generalization

In this section, we develop the CORAL+SDG and VREX+SDG method considering an arbitrary penalty function. To apply the derivation in Section C.1, we need to compute $\texttt{Penalty}(\theta^t + G_k; \{x^i, e^i, y^i\})$ for an arbitrary $G^k$. We consider the first order approximation of this penalty as,

$$\texttt{Penalty}(\theta^t + G_k; \{x^i, e^i, y^i\}) \approx \texttt{Penalty}(\theta^t; \{x^i, e^i, y^i\}) + \nabla_\theta \texttt{Penalty}(\theta^t; \{x^i, e^i, y^i\})^\intercal G_k \tag{27}$$

By this approximation, we do not need to perform additional computations for $\texttt{Penalty}$ for each $G_k$, instead we can compute the derivative once and perform dot-products for the rest. We substitute this approximation in the Blahut-Arimoto iteration eventually obtaining;

- $\hat{p}^{l+1}(G_k|E=e) = p^l(G_k)\exp\left[-\frac{1}{\gamma}\left(\beta d(G_k,e) + \nabla_\theta \texttt{Penalty}(\theta^t)^\intercal G_k\right)\right]$

- $p^{l+1}(G_k|E=e) = \frac{\hat{p}^{l+1}(G_k|E=e)}{\sum_{\hat{k}}\hat{p}^{l+1}(G_{\hat{k}}|E=e)}$

- $p^{l+1}(G_k) = \frac{1}{|E|}\sum_{E=e}p^{l+1}(G_k|E=e)$

We further apply two approximations, i) solving each parameter independently, and ii) solving only for the direction of the gradient (positive or negative). When these approximations are applied, the resulting iterations can be efficiently performed via vector operations and summarized in Algorithm 1.

---

**Algorithm 1** PenaltyBasedDeepDG+SDG-Single Iteration

---

$G_+ = 0, G_- = 0$          ▷ Initialize positive grad, and negative grad with $0$ Vector
$G_{penalty} = \nabla_\theta L^{penalty}(\theta^t)$          ▷ Gradient of the penalty
**for** $e \in \{1,\dots,|E|\}$ **do**          ▷ For each domain
    $G_e = \frac{1}{|E|}\nabla_\theta \mathcal{L}^e(\theta^t)$
    $G_+ = G_+ + \mathbb{1}[G_e > 0]\cdot G_e$          ▷ Parameters with positive gradients
    $G_- = G_- + \mathbb{1}[G_e < 0]\cdot G_e$          ▷ Parameters with negative gradients
**end for**
$Prob_+ = \textsc{BlahutArimotoStyleSolver}(G_1,\dots,G_{|E|},G_{penalty})$
$P_+ \sim Bern(Prob_+)$          ▷ Sample the gradient directions
$G = P_+\cdot G_+ + (1-P_+)\cdot G_-$
$\theta^{t+1} = \theta^t - \eta G$
**procedure** $\textsc{BlahutArimotoStyleSolver}(G_1,\dots,G_{|E|},G_{penalty})$
    **Initialize:** $Prob_+ = Prob_- = 0.5$          ▷ Initialize probabilities uniformly
    **for** $iter = 1,\dots,IterCount$ **do**
        **for** $e \in \{1,\dots,|E|\}$ **do**          ▷ All Operations are Elementwise during the Loop
            $\tilde{Prob}_{+,e} = Prob_+ \cdot \exp\left(-\frac{1}{\gamma}\left[\beta(G_e - G_+)\cdot(G_e - G_+) + G_{penalty}\cdot G_+\right]\right)$
            $\tilde{Prob}_{-,e} = Prob_- \cdot \exp\left(-\frac{1}{\gamma}\left[\beta(G_e - G_+)\cdot(G_e - G_+) + G_{penalty}\cdot G_+\right]\right)$
            $Prob_{+,e} = \tilde{Prob}_{+,e}/(\tilde{Prob}_{+,e} + \tilde{Prob}_{-,e})$
            $Prob_{-,e} = 1 - Prob_{+,e}$
        **end for**
        $Prob_+ = \frac{1}{|E|}\sum_e Prob_{+,e}$
        $Prob_- = 1 - Prob_+$
    **end for**
    **return** $Prob_+$
**end procedure**

---

### C.3 Applying Blahut-Arimoto Style Method to Fish [39]

Fish [39] is not using a direct penalty function; hence, applying SDG requires some modifications. In order to apply the SDG to Fish, we simply use the approximation provided in the original work [39]. Specifically, Shi et al. [39] shows that when an inner update of SGD is applied to $\theta^t$ for a few domains to obtain $\tilde{\theta}^t$, $\tilde{\theta}^t - \theta^t$ approximates the gradient of the penalty. Hence, we utilize this fact to perform the iterations as

- $\hat{p}^{l+1}(G_k|E=e) = p^l(G_k)\exp\left[-\frac{1}{\gamma}\left(\beta d(G_k,e) + (\tilde{\theta}^t - \theta)^\intercal G_k\right)\right]$

- $p^{l+1}(G_k|E=e) = \frac{\hat{p}^{l+1}(G_k|E=e)}{\sum_{\hat{k}}\hat{p}^{l+1}(G_{\hat{k}}|E=e)}$

- $p^{l+1}(G_k) = \frac{1}{|E|}\sum_{E=e}p^{l+1}(G_k|E=e)$

## D   Generalization Gap in WILDS [23]

We argued that when the empirical estimation error of the invariance penalty is high, the resulting optimization problem might hurt empirical risk (in-distribution performance) and fails to perform out-of-distribution even if it generalizes well. To further quantify this motivation, we look at the

benchmarking results of [23]. We specifically enclosed the out-of-distribution and in-distribution results of various benchmarked algorithms from [23] in Table 6&7. Moreover, we compute the generalization gap as the difference between in-distribution and out-of-distribution performance and tabulate them in Table 8.

The results confirm the failure mode we hypothesized. Specifically, ERM outperforms existing domain generalization algorithms in almost all benchmarks when the metric is out-of-distribution performance. However, existing domain generalization methods generalize better (generalization gap is smaller than ERM). Hence, the failure of these methods is not due to the lack of generalization but due to the lack of effective minimization of the empirical risk.

Table 6: **Out-of-distribution test performance** of the benchmarked algorithm reported by [23]. ERM overwhelmingly outperforms existing domain generalization algorithms.

| DATASET | METRIC | ERM | CORAL | IRM | GROUPDRO |
|---------|--------|-----|-------|-----|----------|
| iWILDCAM | MACRO F1 | $31.0 \pm 1.3$ | $\mathbf{32.8 \pm 0.1}$ | $15.1 \pm 4.9$ | $23.9 \pm 2.1$ |
| CAMELYON17 | AVERAGE ACC. | $\mathbf{70.3 \pm 6.4}$ | $59.5 \pm 7.7$ | $64.2 \pm 8.1$ | $68.4 \pm 7.3$ |
| RxRx1 | AVERAGE ACC. | $\mathbf{29.9 \pm 0.4}$ | $28.4 \pm 0.3$ | $8.2 \pm 1.1$ | $23.0 \pm 0.3$ |
| OGBMOLPCBA | AVERAGE AP | $\mathbf{27.2 \pm 0.3}$ | $17.9 \pm 0.5$ | $15.6 \pm 0.3$ | $22.4 \pm 0.6$ |
| FMoW | WORST REGION ACC. | $\mathbf{32.3 \pm 1.3}$ | $31.7 \pm 1.2$ | $30.0 \pm 1.4$ | $30.8 \pm 0.8$ |
| POVERTYMAP | WORST REGION PEARSON R | $\mathbf{0.45 \pm 0.06}$ | $0.44 \pm 0.06$ | $0.43 \pm 0.07$ | $0.39 \pm 0.06$ |
| AMAZON | 10TH PERCENTILE ACC. | $\mathbf{53.8 \pm 0.8}$ | $52.9 \pm 0.8$ | $52.4 \pm 0.8$ | $53.3 \pm 0.1$ |
| PY150 | METHOD/CLASS ACC | $\mathbf{67.9 \pm 0.1}$ | $65.9 \pm 0.1$ | $64.3 \pm 0.2$ | $65.9 \pm 0.1$ |

Table 7: **In-distribution test performance** of the benchmarked algorithm reported by [23]. ERM overwhelmingly outperforms existing domain generalization algorithms. Hence, domain generalization algorithms struggle to minimize empirical risk.

| DATASET | METRIC | ERM | CORAL | IRM | GROUPDRO |
|---------|--------|-----|-------|-----|----------|
| iWILDCAM | MACRO F1 | $\mathbf{47.0 \pm 1.4}$ | $43.5 \pm 3.5$ | $22.4 \pm 7.7$ | $37.5 \pm 1.7$ |
| CAMELYON17 | AVERAGE ACC. | $93.2 \pm 5.2$ | $\mathbf{95.4 \pm 3.6}$ | $91.6 \pm 7.7$ | $93.7 \pm 5.2$ |
| RxRx1 | AVERAGE ACC. | $\mathbf{35.9 \pm 0.4}$ | $34.0 \pm 0.3$ | $9.9 \pm 1.4$ | $28.1 \pm 0.3$ |
| OGBMOLPCBA | AVERAGE AP | $\mathbf{27.8 \pm 0.1}$ | $18.4 \pm 0.2$ | $15.8 \pm 0.2$ | $23.1 \pm 0.6$ |
| CIVILCOMMENTS | WORST GROUP ACC | $50.5 \pm 1.9$ | $64.7 \pm 1.4$ | $65.9 \pm 2.8$ | $\mathbf{67.7 \pm 1.8}$ |
| FMoW | WORST REGION ACC. | $\mathbf{58.30 \pm 0.92}$ | $55.00 \pm 1.02$ | $56.00 \pm 0.34$ | $57.80 \pm 0.60$ |
| POVERTYMAP | WORST REGION PEARSON R | $0.57 \pm 0.07$ | $\mathbf{0.59 \pm 0.03}$ | $0.57 \pm 0.08$ | $0.54 \pm 0.11$ |
| AMAZON | 10TH PERCENTILE ACC. | $\mathbf{57.3}$ | $55.1 \pm 0.4$ | $54.7 \pm 0.1$ | $55.8 \pm 1.0$ |
| PY150 | METHOD/CLASS ACC | $\mathbf{75.4 \pm 0.4}$ | $70.6$ | $67.3 \pm 1.1$ | $70.8$ |

Table 8: **Generalization Gap** (difference between in-distribution and out-distribution performance) in the WILDS benchmarks [23]. We denote the results which generalize better than ERM with Green color and worse than ERM with Red color. Domain generalization algorithms typically generalize better than ERM. Hence, although they are good generalizers, they fail to perform out-of-distribution due to poor in-distribution performance.

| DATASET | METRIC | ERM | CORAL | IRM | GROUPDRO |
|---------|--------|-----|-------|-----|----------|
| iWILDCAM | MACRO F1 | 16 | 10.7 | **7.3** | 13.6 |
| CAMELYON17 | AVERAGE ACC. | **22.9** | 35.9 | 27.4 | 25.3 |
| RxRx1 | AVERAGE ACC. | 6 | 5.6 | **1.7** | 5.1 |
| CIVILCOMMENTS | WORST GROUP ACC | 5.5 | 0.9 | **0.4** | 2.3 |
| OGBMOLPCBA | AVERAGE AP | 0.6 | 0.5 | **0.2** | 0.7 |
| FMoW | WORST REGION ACC. | 26 | **23.3** | 26 | 27 |
| POVERTYMAP | WORST REGION PEARSON R | **0.12** | 0.15 | 0.14 | 0.15 |
| AMAZON | 10TH PERCENTILE ACC. | 3.5 | **2.2** | 2.3 | 2.5 |
| PY150 | METHOD/CLASS ACC | 7.5 | 4.7 | **3** | 4.9 |

# E  Details on Used Datasets

**iWildCam** [8]: The problem is predicting animal species from camera trap images over different locations. The dataset includes 323 different camera traps, considered as domains. Among these, 243 are used for training, 32 for validation, and 48 for testing. The evaluation metric is the F1-scores.

**Camelyon17** [7]: The problem is predicting tumors from tissue slides over 5 different hospitals, treated as domains. Among these, 3 are used as training, 1 for validation, and 1 for test. The metric is the accuracy over balanced patches (i.e. same number of positive and negative patches).

**OGB-MolPCBA** [21]: The problem is predicting bioassays from molecular graphs over 120,084 scaffolds, considered as domains. The largest 44,390 scaffolds are used for training, the next largest 31,361 are for validation, and the smallest 43,793 are for testing. The metric is average precision.

**Functional Map of the World (FMoW)** [12]: The problem is predicting the land usage from a satellite image from 5 regions over 16 years, a total of $5 \times 16 = 80$ domains. From 16 years of data, 11 years are used for training, 3 for validation, and 2 for testing. The evaluation metric is the accuracy over the worst region to evaluate the sub-population shift.

**Amazon Reviews** [29]: The problem is predicting sentiment from product reviews over 3920 different users considered as domains. Among those, 1252 users are used for training, 1334 for validation, and 1334 for testing. The evaluation metrics are accuracy averaged over users in the 10th percentile.

**py150** [33]: The problem is autocompletion of Python code over 8421 git repositories as domains. Among them, 5477 are used for training, 261 for validation, and 2471 for testing. The evaluation is the accuracy of predicting class/method name tokens.

**PovertyMap** [47]: The problem is predicting asset wealth from satellite images over different countries. Evaluation is over 5 different folds where 13-14 countries are used for training, 4-5 for validation, and 4-5 for the test. The metric is the Pearson correlation averaged over the rural regions.

# F  Hyper Parameters

We share the implementation of our method and experimental setup for all WILDS experiments. In order to reproduce the experiments, the scripts can be run with the following hyperparameters. Any hyperparameter not shared here, directly follows the default one shared by [23]. For DomainBed experiments, we directly used the recommended hyperparameter search pipeline without any modifications.

We perform a hyperparameter search for the WILDS experiments with the same budget reported in [23]. The chosen hyperparameters are given in Table 9.

Table 9: Chosen hyperparameters for the WILDS [23] experiments.

|  | Learning Rate | Batch Size | N Groups per Batch |
|---|---|---|---|
| iWildCam | $3 \times 10^{-5}$ | 32 | 4 |
| Camelyon17 | $5 \times 10^{-3}$ | 60 | 3 |
| OGB-MolPCBA | $1 \times 10^{-3}$ | 64 | 32 |
| FMoW | $1 \times 10^{-4}$ | 32 | 4 |
| Amazon | $1 \times 10^{-5}$ | 32 | 8 |
| Py150 | $1 \times 10^{-5}$ | 6 | 2 |
| PovertyMap | $5 \times 10^{-4}$ | 64 | 4 |

# G   Societal Impact

Our work studies the optimization of domain generalization algorithms by re-formulating the joint optimization problem of domain generalization as constrained optimization of penalty under the constraint of optimality of the empirical risk. Our approach is purely algorithmic and applicable to any penalty-based method. Hence, it does not change the societal impact of using the original method and dataset. It is still beneficial to re-iterate some aspects of the societal impact of the state of the domain generalization algorithms. Our work, among others, shows that immediate deployment of machine learning models without investigating the fairness and safety when applied to the target distribution might not be feasible. Performance and behavior on unseen distributions are still far from being predictable. Hence, particular care should be spent studying and monitoring the target distribution and its changes over time.

Moreover, a critical aspect of societal impact is the datasets. Fortunately, WILDS benchmark [23] provides a comprehensive analysis for the datasets we use. We refer the interested reader to this comprehensive analysis.