# OpenReview forum: "Domain Generalization without Excess Empirical Risk"
_NeurIPS.cc/2022/Conference — NeurIPS 2022 Accept_

### Official Review · Reviewer_zhHr · 2022-07-10

**Rating:** 5
**Confidence:** 3
**Soundness:** 2 fair
**Presentation:** 3 good
**Contribution:** 3 good

**Summary:**

This paper argues that the joint optimization of the empirical risk and the penalty in penalty-based DG methods can increase empirical risk, which hurts performance on both training and unseen distributions. Based on this hypothesis, this paper proposes to minimize the penalty under the constraint of optimality of the empirical risk and presents solutions by connecting to rate-distortion theory. Experiments show that the proposed optimizer (SDG) effectively improves the OOD test performance.

**Questions:**

My main concerns are about the evaluation of experiments. Please see weaknesses for more details.

**Limitations:**

The authors addressed the limitations of their work. I didn't see potential negative societal impact of this work.

**Strengths And Weaknesses:**

Strengths (originality, quality, clarity and significance):
1. As far as I know, the originality is good.
2. The proposed method is theoretically and technically sound.
3. This paper is well organized.
4. The obversion of the generalization gap and in-distribution performance in the WILDS benchmarks is interesting. The proposed optimizer (SDG) significantly improves the performances of CORAL, FISH, and VREx on several datasets.

Weaknesses:
1. The evaluation of the proposed optimizer (SDG) is a little weak. Experiments with more penalty-based DG methods will make the effectiveness of SDG stronger.
2. The statement that the proposed optimizer improves empirical risk without hurting the generalization ability as desired in lines 184 – 287 is unconvincing. In Table 2, SDG increases the generalization gap on four datasets (IWILDCAM., OGBMOLPCBA, AMAZON, and PY150) and decreases the generalization gap on only three datasets (CAMELYON17, FMoW, and POVERTYMAP).
3. In lines 309 – 311, the explanation that the performance of the FISH for FMoW is already strong is far-fetched with the worst accuracy of 34.6.

---

> ### Author Response · Authors · 2022-08-01
> **Answering the Review Questions**
>
> We thank the reviewer for their insightful comments and questions. We answer them as follows:
>
> ---
>
> > The evaluation of the proposed optimizer (SDG) is a little weak. Experiments with more penalty-based DG methods will make the effectiveness of SDG stronger.
>
> We believe the coverage of the algorithms is extensive. We considered three (CORAL, FISH, and VRex) different DG methods and showed effectiveness on two (WILDS and DomainBed) benchmarks. Since the submission, we also evaluated CORAL and FISH on DomainBed. Following other reviews, we also evaluated FISH on the remaining WILDS datasets. We believe these additional experiments complete the experimental study and consistently demonstrate the effectiveness of our method.
>
> ---
>
> > The statement that the proposed optimizer improves empirical risk without hurting the generalization ability as desired in lines 184 – 287 is unconvincing. In Table 2, SDG increases the generalization gap on four datasets (IWILDCAM., OGBMOLPCBA, AMAZON, and PY150) and decreases the generalization gap on only three datasets (CAMELYON17, FMoW, and POVERTYMAP).
>
> We clarified the claim in the revised paper. Among the four datasets where the generalization gap is increased, for two (IWILDCAM and AMAZON), the difference is within the std-dev of the reported in-distribution and out-of-distribution performances. Hence, our method does not hurt generalization in 5 out of 7 datasets. Moreover, it improves empirical risk in ALL datasets, as shown in Table 9 in Appendix. Although the statement does not uniformly hold over all datasets as suggested by the reviewer, it is valid when considered over all the datasets. We thank the reviewer for raising the inconsistency and hope the revised version is convincing.
>
> ---
>
> > In lines 309 – 311, the explanation that the performance of the FISH for FMoW is already strong is far-fetched with the worst accuracy of 34.6.
>
> We agree with the reviewer and removed the claim in the revised version.

---

> > ### Comment · Reviewer_zhHr · 2022-08-08
> > **Response to Authors**
> >
> > Thanks for the authors' responses. According to additional experiments, SDG does not consistently improve the evaluated methods, e.g., FISH on FMoW, R-MNIST, Office-Home and DomainNet datasets, CORAL on Office-Home and DomainNet datasets. I think the statement that the proposed method consistently improves the performance of all tested methods (L312-313) is overclaimed. Could the authors give some discussions on these failure examples?

---

> > > ### Author Response · Authors · 2022-08-09
> > > **Addressing the Issues**
> > >
> > > Thanks for the response. We believe there is a slight misunderstanding primarily caused by our ambiguous statement. When we said "... consistently improves the performance of all tested methods...", we meant the improvement is consistent over algorithms, not for every single experiment. Our method improves all algorithms' average performance (the metric used in DomainBed). We will revise the sentence as follows: "According to the average performance reported in Table 3, our method consistently improves the performance of all tested methods."
> > >
> > > About the cases where our method does not change the performance of the tested method: We will add a discussion of why this can be happening. Our method introduces an additional constraint to the domain generalization methods. If this constraint is already satisfied by the original method, our method would not be effective. The geometry presented in Figure 1 explains this behavior. When the penalty minimizing update is already a satisficing update (i.e., the yellow shaded region includes the orange point), our method does not change the behavior (i.e., orange point and red point overlap). In such a case, our method will perform the same as the original method. We will update the text with an explanation of this failure case as well as the underlying geometric explanation.

---

### Official Review · Reviewer_Uqg1 · 2022-07-12

**Rating:** 7
**Confidence:** 3
**Soundness:** 3 good
**Presentation:** 4 excellent
**Contribution:** 3 good

**Summary:**

This work argues that penalty-based DG methods often perform poorly in practice as, due to the hardness of joint optimization, they fail to minimize the empirical risk (i.e. in-distribution performance). To address this, the authors: (1) assume that excess risk is not required for generalization, i.e. that in-domain performance need not be sacrificed for out-of-domain performance; and (2) eliminate excess empirical risk by reformulating penalty-based DG such that the penalty is minimized under the constraint of optimal empirical risk. Then, drawing on connections to rate-distortion theory, a method is proposed to solve the proposed optimization problem. Experiments on the WILDS and DomainBed benchmark datasets illustrate the effectiveness of this method.

**Questions:**

- See above.
- In Eq. 6, adding in another separate penalty/regularization seemed strange to me -- can't this just be absorbed into the abstract $\texttt{Penalty}(\\cdot)$ term? Was this addition purely for the theory/derivations? Does it make a difference empirically (e.g. via an ablation study)?

**Limitations:**

Yes, the authors discuss the computational complexity of their method as a limitation in 3.2.2 and in the conclusion.

**Strengths And Weaknesses:**

**Strengths**
- *Clarity of writing*
    - This paper was a joy to read. Lucid writing, easy to follow, well organised.
    - Sections 2.2--2.4 were particularly impressive.
- *Novel perspective and interesting solution*
    - Blaming excess risk for the failures of penalty-based DG methods seems new, as does formulating a DG problem in which the constraint pertains to the empirical risk rather than the penalty (e.g. as in IRM).
    - While I disagree with the assumptions (see below), I think the perspective, solution and results are valuable to the community.
- *Nice theory and connections to rate-distortion*

**Weaknesses**
- *The reported experimental results seem hand-picked*
    - CORAL and ERM are reported on 6 WILDS datasets, along with their generalization gaps.
    - FISH is reported on 3 WILDS datasets.
    - VREx is reported on DomainBed.
    - The ad-hoc reporting makes me question the results -- why not report all methods on all datasets? In particular, I would question why:
      - ERM is left out of DomainBed
      - CORAL and FISH are left out of DomainBed
      - FISH is only reported on a 3 of 6 WILDS datasets -- this seems to be due to the hyperparameter ranges of FISH being designed/reported for only 3 or 4 datasets, but the results could still be reported to see if SDG improves FISH.
    - Happy to increase my score if this is addressed.
- *The central assumption---in-domain performance need not be sacrificed for out-of-domain performance---is often violated*
    - This essentially assumes no spurious correlations in the (possibly few) training domains---a central motivation for DG methods in the first place, with many works reporting real-world failure modes due to spurious correlations (e.g. [1]).
    - The authors justify this by claiming that:
      - "for the benchmarks we are interested in, which involve supervised learning from multiple datasets, this is not the case as a universal model solving all datasets (seen and unseen) exists" -- here I would question both the existence of such a universal model and its inference from a small subset of the domains.
      - "Since the models we use are significantly overparameterized, the space of solutions to empirical risk minimization is quite large, likely including many domain-invariant and domain-sensitive solutions." -- while I agree that the set of minimal-risk solutions is large, I would question how often domain-invariant solutions lie in this set.


[1] Geirhos, R., Jacobsen, J. H., Michaelis, C., Zemel, R., Brendel, W., Bethge, M., & Wichmann, F. A. (2020). Shortcut learning in deep neural networks. *Nature Machine Intelligence*, 2(11), 665-673.

---

> ### Author Response · Authors · 2022-08-01
> **Answering the Review Questions**
>
> We thank the reviewer for their insightful comments and questions. We answer them as follows:
>
> ---
>
> > The reported experimental results seem hand-picked
>
> In our initial submission, we only considered VRex on DomainBed to simultaneously test applications on a new benchmark and a new algorithm due to the computational cost. Since the submission, we have extended the results to all three methods. We updated the paper with results on CORAL, FISH, VRex, their proposed SDG variants, and the ERM.
>
> For evaluating FISH on WILDS, we initially preferred sticking to the original paper's design space. During the rebuttal period, we performed analysis on the remaining WILDS benchmarks utilizing our own hyperparameter space. We updated the paper with the results of FISH on the entire WILDS benchmark.
>
> We believe these results complete the experimental study. Moreover, the main conclusion of the paper remains.
>
> ---
>
> > The central assumption---in-domain performance need not be sacrificed for out-of-domain performance---is often violated
>
> We largely agree with the reviewer that this is a restrictive setting and a major limitation of our approach. Moreover, we openly discuss this limitation in our paper.
>
> To further address the specific issues raised by the reviewer, we believe the definition of invariance and its numerical optimization can be somewhat contradictory. We consider the two extreme cases of the spectrum as ignoring the domain invariance entirely (as in the case of ERM) with a somewhat easy numerical optimization and a well-defined domain invariance (as in the case of many existing DG methods) with a somewhat hard numerical optimization problem. In this paper, we choose to forgo the perfect definition of domain invariance without an effective numerical optimizer for an imperfect definition enabling better optimization. In other words, we try to control the tradeoff between modeling accuracy and optimization ease.
>
> ---
>
> > In Eq. 6, adding in another separate penalty/regularization seemed strange to me -- can't this just be absorbed into the abstract  term? Was this addition purely for the theory/derivations? Does it make a difference empirically (e.g. via an ablation study)?
>
> We assume Penalty term is given by the definition of the domain invariance objective. For example, for CORAL, it is the penalty term of CORAL, and for FISH, it is the penalty term corresponding to FISH. We envision our method as an add-on to an existing domain generalization method. In other words, we have no control over the Penalty() term.
>
> We need this term to derive the proposed optimizer as the Blahut-Arimato derivation depends on it. To the best of our knowledge, we are unaware of a rate-distortion solver, which would not work without such a term.

---

> > ### Comment · Reviewer_Uqg1 · 2022-08-04
> > **Response**
> >
> > The authors have addressed my main concern regarding the empirical evaluations. I have thus increased my score, as promised.

---

### Official Review · Reviewer_AMTZ · 2022-07-15

**Rating:** 6
**Confidence:** 3
**Soundness:** 3 good
**Presentation:** 3 good
**Contribution:** 3 good

**Summary:**

This paper argues that domain generalization methods that jointly minimize the empirical risk with a penalty term fail to decrease the empirical risk and hurt the performance both on the source and unseen target domains. Therefore, it proposes an optimization algorithm that minimizes the penalty under the constraint of optimality of the empirical risk.

**Questions:**

- Should be \hat{L} in Eq. (4), (5), and (6) be L since L is already the empirical risk defined in Line 87?
- Why is the penalty minimized at \theta^t + G^t in Eq. (5) and (6)? Why is it not \theta^t - \eta * G^t?
- It would be better to point out in the paper instead of the appendix that the constraint is replaced with a stronger constraint of uniform bounds over domains (Line 602, Appendix).

**Limitations:**

See weaknesses.

**Strengths And Weaknesses:**

Strengths:
- The proposed algorithm in this paper provides a new idea for manipulating the gradients during the training for penalty-based domain generalization methods.
- Overall, the paper is well written and organized.

Weaknesses:
- As the authors point out, the proposed method can only solve domain generalization problems where a universal model solving all domains (seen and unseen) exists, which could be a weakness when applied to real-world applications.
- The proposed simplification for the space of gradients for updating looks over-simplified. It would be better to provide some justifications for the choice of gradient space in the paper or consider more sophisticated choices.

---

> ### Author Response · Authors · 2022-08-01
> **Answering the Review Questions**
>
> We thank the reviewer for their insightful comments and questions. We answer them as follows:
>
> ---
>
> > The proposed simplification for the space of gradients for updating looks over-simplified
>
> We updated the paper to motivate the proposed simplification. We included the following paragraph in L231-238:
>
> To understand the proposed simplifications, we consider their implications for the rest of the pipeline. We feed the resulting estimated gradients to first-order numerical optimizers. Moreover, we utilize the first-order approximation of the penalty function in our algorithms in Appendix C1 and C2. Hence, the rest of the pipeline is first order, not utilizing higher-order relationships between coordinates, justifying the first simplification. The second simplification is similar to SignSGD [10] like methods, where only the sign of the gradient is used without its magnitude for efficient communication between worker nodes in multi-node optimization. Sign of the gradient suffices for convergence in non-convex problems both theoretically and empirically.
>
> ---
>
> > Should be \hat{L} in Eq. (4), (5), and (6) be L since L is already the empirical risk defined in Line 87?
>
> Thanks for catching this typo. \hat{L} should be L. We fixed it in the revised version.
>
> ---
>
> > Why is the penalty minimized at \theta^t + G^t in Eq. (5) and (6)? Why is it not \theta^t - \eta * G^t?
>
> We preferred to use G^t since we consider the output of our method as a replacement for a gradient which is further utilized by a numerical optimization method. The step size (\eta) is decided by the numerical optimizer and can be unknown to our method. For example, adaptive optimizers set the step size after the gradient is revealed.
>
> ---
>
> > It would be better to point out in the paper instead of the appendix that the constraint is replaced with a stronger constraint of uniform bounds over domains (Line 602, Appendix).
>
> Thanks for the suggestion. We agree and modified the paper accordingly (L214-217).

---

### Meta-Review · Area_Chair_xFAT · 2022-08-30

**Recommendation:** Accept
**Confidence:** Certain

**Metareview:**

The paper considers the class of penalty-based methods methods for domain generalization which aim at minimizing a combination of the empirical risk and a penalty term used as proxy of the generalization error in the unseen domains (examples include IRM, CORAL, etc). The paper argues that such methods often perform poorly in practice either due to an erroneous penalty term, or due to the hardness of joint optimization which results in the failure to minimize the empirical risk. To address this issue,  the authors propose a method which, instead of jointly minimizing empirical risk with the penalty, it minimizes the penalty under the constraint of optimality of the empirical risk. The success of this methods is based on the assumption that assume that excess risk is not required for generalization. In other words,  in-domain performance need not be sacrificed for out-of-domain performance (no trade-off between the two). As a result, the proposed method essentially eliminates excess empirical risk by reformulating penalty-based DG such that the penalty is minimized under the constraint of optimal empirical risk.

All the three reviewers found the results novel an interesting. The reviewers had some concerns (about the framework and the experiments) which were addressed during the discussion period. I recommend that the authors also apply all the reviewers' comments in the revised manuscript. Also, for the experiments, I think it'd useful to compare the numbers with SOTA numbers for the datasets considered in the experiments (they are all available in the WILDS leaderboard).



**Award:**

No

---

### Decision · Program_Chairs · 2022-09-14

Accept